# A Degradation Warning Method for Ultra-High Voltage Energy Devices Based on Time-Frequency Feature Prediction

**DOI:** 10.3390/s25113478

**Published:** 2025-05-31

**Authors:** Pinzhang Zhao, Lihui Wang, Jian Wei, Yifan Wang, Haifeng Wu

**Affiliations:** 1Jiangsu Institute of Metrology, Nanjing 210023, China; lemon_jsmi@163.com (P.Z.); 13645162540@163.com (J.W.); 2Key Laboratory of Micro-Inertial Instrument and Advanced Navigation Technology, Ministry of Education, School of Instrument Science and Engineering, Southeast University, Nanjing 210096, China; wangyf0002@163.com (Y.W.); wuhaifeng199907@163.com (H.W.)

**Keywords:** ultra-high voltage energy device, fault warning, leakage current, signal decomposition, feature prediction

## Abstract

This study addresses the issue of resistance plate deterioration in ultra-high voltage energy devices by proposing an improved symplectic geometric mode decomposition-wavelet packet (ISGMD-WP) algorithm that effectively extracts the component characteristics of leakage currents. The extracted features are subsequently input into the I-Informer network, allowing for the prediction of future trends and the provision of early short-term warnings. First, we enhance the symplectic geometric mode decomposition (SGMD) algorithm and introduce wavelet packet decomposition reconstruction before recombination, successfully isolating the prominent harmonics of leakage current. Second, we develop an advanced I-Informer prediction network featuring improvements in both the embedding and distillation layers to accurately forecast future changes in DC characteristics. Finally, leveraging the prediction results from multiple adjacent columns mitigates the impact of power grid fluctuations. By integrating these data with the deterioration interval, we can issue timely warnings regarding the condition of lightning arresters across each column. Experimental results demonstrate that the proposed ISGMD-WP effectively decomposes leakage current, achieving a decomposition ability evaluation index (EIDC) 1.95 under intense noise. Furthermore, in long-term prediction, the I-Informer network yields mean absolute error (MAE) and root mean square error (RMSE) indices of 0.02538 and 0.03175, respectively, enabling the accurate prediction of the energy device’s fault.

## 1. Introduction

With advancements in large capacity, long-distance, and direct current (DC) transmission, the power system is transitioning to ultra-high voltage. A hybrid DC power grid has already been established [1]. Recently, a controllable self-recovery energy dissipation device has been developed to effectively limit pressure and release energy at both ends of the converter valve. This was implemented within a hybrid flexible DC engineering converter station [2]. The core of this controllable self-recovery energy dissipation device is a series-parallel structure consisting of hundreds of lightning arresters. During actual operation, parameter variances and volt-ampere characteristics among these resistors can occur [3]. Poor sealing of the lightning arrester unit can lead to moisture intrusion into the resistor [4], while dirt accumulation on the outer surface of the lightning arrester can impair performance [5]. These challenges result in uneven voltage and current distribution of multi-column series-parallel DC lightning arresters, further complicating the harmonics of the DC system [6]. The stray capacitance will further exacerbate this phenomenon, and the inherent characteristics of lightning arresters may lead to their degradation. Long-term uneven potential distribution caused by the external environment further exacerbates this issue. Additionally, exposure to operational overvoltage and lightning overvoltage energy impact can intensify the degradation of some resistance pieces, prompting the formation of short pieces [7]. These short pieces can increase the voltage and current load on other resistance pieces in different positions, adversely affecting the overall lifespan of the entire lightning arrester and potentially jeopardizing the operational safety of the arrester itself and other electrical equipment.

Currently, methods of monitoring lightning arresters are categorized based on their detection techniques: electromagnetic radiation detection method, infrared temperature detection method, and leakage current detection. The electromagnetic radiation detection method captures partial discharge signals from surface defects of lightning arresters using high-frequency electromagnetic radiation [8]. However, this approach cannot detect internal deterioration and faults in lightning arresters and is vulnerable to substantial electromagnetic interference in environments such as converter stations. Infrared temperature detection identifies local temperature anomalies attributed to faults or deterioration in the arresters through imaging technology [9]. Although unaffected by electromagnetic environment and power grid fluctuations, this method suffers when faced with environmental factors such as temperature and weather, particularly in complex backgrounds and structures such as controllable self-recovery energy dissipation devices.

The leakage current detection method can be subdivided into complete and resistive current methods, depending on their focuses. The whole current method measures the total current without differentiating between the resistive and capacitive currents in the lightning arrester. One study employs a detector to gather complete current information from the lightning arrester for temporal and phase comparisons to monitor the operational state [10]. Their findings reveal that the total current change primarily arises from changes in resistive current, even though the resistive current is a small component of the total current. However, this method lacks accuracy when total current fluctuations are minimal, even though changes primarily indicate variations in resistive current. The resistive current rule extracts the resistive current from the total current for analysis. Various extraction methods exist, such as monitoring external voltage and electric field signals to adjust for capacitive currents [11], although this approach raises equipment costs. By taking advantage of the 90-degree phase relationships of capacitive current over resistive current, researchers can manipulate the total current signal to eliminate capacitive current [12]. However, high-order harmonic signals can distort these results.

Additionally, using a three-phase current superposition can eliminate fundamental waves and extract the third harmonic of the resistive current for analysis [13]. However, this approach is unsuitable for non-three-phase circuits. Furthermore, it cannot determine the phase of the lightning arrester. Fast Fourier transform can provide current harmonic amplitude and frequency information [14], but the accuracy can be compromised by spectral leakage and fence effects. Emerging techniques utilizing signal decomposition algorithms and machine learning are increasingly applied for feature extraction and lightning arrester fault assessment. One study applied variational mode decomposition to denoise leakage current signals before estimating the aging parameters of lightning arresters through a particle swarm optimization algorithm for online monitoring [15]. In recent years, the end-to-end deep learning method has shown potential in leakage current analysis. The long-term and short-term memory convolution network (LSTM-CNN) effectively captures the long-term patterns in sequential data through LSTM, while the CNN layer effectively extracts the high-level dependencies in time invariant information, which effectively improves the robustness and accuracy of leakage current classification [16,17]. In addition, the attention mechanism is also applied to the prediction of leakage current, and the performance and stability of the prediction model are improved by providing the weighted sum of the hidden states of LSTM units [18].

The Symplectic Geometric Mode Decomposition (SGMD) algorithm has gained significant attention in signal processing due to its superior decomposition capabilities and its ability to preserve time-frequency characteristics. Nonetheless, the SGMD algorithm encounters practical challenges, such as choosing embedding dimensions, reconstructing initial component matrices, and determining the recombination strategy. These factors can significantly limit the effectiveness and applicability of decomposition.

This study proposes an improved symplectic geometric mode decomposition-wavelet packet (ISGMD-WP) algorithm, explicitly separating leakage current components to address SGMD’s shortcomings. Additionally, we introduced an I-Informer network aimed at predicting component features and issuing warnings for short circuits.

This article’s main contributions include several significant advancements:

First, it enhances the embedding dimension determination, initial single-component matrix reconstruction, component recombination, and other related steps of SGMD.

Second, the article introduces wavelet packet decomposition reconstruction to improve the algorithm’s noise robustness before recombining the SGMD components.

Third, it establishes an I-Informer network to achieve low prediction error for long-term prediction of DC features.

Lastly, the article combines the predicted values of the adjacent multi-column DC characteristics to minimize voltage fluctuation interference in the power grid and achieve an early warning of lightning arrester short section status.

## 2. Leakage Current Harmonic Separation Algorithm

The leakage current components can be separated using signal decomposition algorithms. Empirical Mode Decomposition (EMD) and Wavelet Packet Decomposition (WP) are widely used signal decomposition methods. Notably, Pan introduced the symplectic geometric modal decomposition algorithm, which was grounded in the analysis of symplectic geometric spectra [19]. This approach facilitates the feature decomposition and fault diagnosis of bearing fault signals. The SGMD algorithm boasts several advantages: it exhibits strong decomposition ability, preserves the essential features of time series unchanged, and suppresses mode confusion. As a result, it is gradually being applied in mechanical fault diagnosis and power signal decomposition.

The SGMD achieves precise signal decomposition through phase space reconstruction and eigenvalue decomposition, while wavelet packets refine signal features via multiresolution analysis. The SGMD technique leverages symplectic geometric analysis and employs power spectral analysis to obtain the energy information of local frequencies. It extracts these components by sorting them based on energy. At its core, the SGMD utilizes symplectic geometric similarity transformations to transform trajectory matrices, solving the eigenvalues of the Hamiltonian matrix and reconstructing Symplectic Geometric Components (SGCs) using corresponding eigenvectors. This transformation allows the adaptive decomposition of complex signals into multiple SGCs, which are then selectively reconstructed based on predetermined criteria.

The SGMD workflow is summarized in four steps: phase space reconstruction, symplectic geometric matrix similarity transformation, diagonal averaging, and component reconstruction. However, this algorithm has limitations, particularly in selecting the embedding dimension during phase space reconstruction and the noise robustness during component reconstruction, which manifests as follows:
The chosen embedding dimension significantly affects results during phase space reconstruction. However, current methods for determining this dimension lack precision.A sizeable embedding dimension may result in processing many ineffective components, leading to time consumption.The methods for initial single-component recombination and the criteria for iteration termination in symplectic geometry are unclear.The algorithm’s ability to suppress noise is insufficient when faced with strong noise signals.

To address these challenges, we propose an improved algorithm for harmonic extraction utilizing a symplectic geometric mode decomposition wavelet packet. The improved method includes the transformation and reconstruction of symplectic geometric matrices. Given that the SGMD can decompose the DC component of the signal into a specific component and mix a particular low-frequency component, resulting in mode aliasing, it is essential to subtract the signal mean before subsequent processing for complete signal centralization.

### 2.1. Symplectic Geometric Matrix Transformation

To address the limitations of the original SGMD algorithm in embedding dimension determination and single-component matrix reconstruction, we introduce an adaptive embedding dimension selection method based on the variance of reconstructed components, ensuring a dynamic and data-driven selection approach. Furthermore, we propose a constrained eigenvalue analysis technique for the initial matrix reconstruction, which effectively suppresses noise components and improves the accuracy of mode decomposition. These improvements are justified through theoretical analysis and prior experimental findings, demonstrating enhanced component separation under varying noise levels.

A multidimensional time series trajectory matrix X of one-dimensional time series $x=\{x_{1},x_{2},\ldots ,x_{n}\}$ is constructed following the Takens embedding theorem, as shown in (1):(1)$$X=\left[\begin{array}{cccc} x_{1} & x_{1+\tau } & \ldots & x_{1+(d-1)\tau }\\ x_{2} & x_{2+\tau } & \ldots & x_{2+(d-1)\tau }\\ \vdots & \vdots & \ddots & \vdots \\ x_{m} & x_{m+\tau } & \ldots & x_{m+(d-1)\tau } \end{array}\right]$$
where *d* is the embedding dimension; τ signifies the delay time, *m* = *n* − (*d* − 1) *τ*. After determining the delay time, the original method used the peak power spectral density to determine the embedding dimension. However, this approach was inaccurate when the signal’s main and sampling frequencies differed significantly. Additionally, it required manual adjustment of threshold and magnification parameters, defined as follows:(2)$$a(i,d)=\frac{\left\|x_{i}(d+1)-x_{j(i,d)}(d+1)\right\|}{\left\|x_{i}(d)-x_{j(i,d)}(d)\right\|}$$
where $x_{i}(d)$ is the ith row vector of the trajectory matrix constructed with embedding dimension *d*; $x_{j(i,d)}(d)$ refers to the nearest neighbor row vector of the vector $x_{i}(d)$; and the nearest neighbor judgment method is the Chebyshev distance. Moreover, $\left\|*\right\|$ signifies the maximum norm calculation, defined as follows:(3)$$E(d)=\frac{1}{N-d\tau }\sum_{i=1}^{N-d\tau }a(i,d)$$(4)$$E1(d)=\frac{E(d+1)}{E(d)}$$
when the delay time τ is determined, *E*1(*d*) is only related to the embedding dimension *d*, and the fluctuation increases as d rises, eventually stabilizing around 1. The initial stable point is identified by simply setting a threshold to determine the minimum embedding dimension *d* when *E*1(*d*) starts to stabilize, which is still not sufficiently precise and lacks robustness when applied to different types of data or strong noise data, defined as follows:(5)$$\delta_{i}=\frac{1}{3}\sum_{j=i}^{i+2}(E1(j)-E1(j-3))$$
where $d_{0}$ is the value of *d* when E1(d) first enters the set threshold range based on the following $i=d_{0},d_{0}+1,\ldots$.

When $\delta_{i}<0$, i determines the initial stable point, and the embedding dimension d is the initial stable point value plus 3. After constructing the trajectory matrix based on the embedding dimension and delay time, we calculate the covariance matrix *A* = *X^T^X* and construct the Hamiltonian matrix:(6)$$M=\left[\begin{array}{cc} A & 0\\ 0 & -A^{T} \end{array}\right]$$

However, the matrix M is insufficient to obtain the symplectic geometry matrix. Further processing of the matrix M yields another Hamiltonian matrix ***N*** = ***M***^2^. After constructing a symplectic orthogonal matrix P to perform a symplectic geometric similarity transformation on *N*, we obtain the following expression:(7)$$P^{T}NP=\left[\begin{array}{cc} R & B\\ 0 & R^{T} \end{array}\right]$$
where R is the upper Hessenberg matrix, and the matrix P is used to construct the Household matrix *Q* through matrix *N*, obtaining the Household matrix H as follows:(8)$$H=\left[\begin{array}{cc} Q & 0\\ 0 & Q \end{array}\right]$$

### 2.2. Reconstruction of Symplectic Geometric Components

The symplectic geometric matrix transformation obtains the symplectic geometric feature matrix *Q* and the upper Hessenberg matrix *R*, proving that the matrix *R* has the same eigenvalues as matrix a, and it is the main diagonal element of the matrix *R*. The column vector corresponding to matrix *Q* is the eigenvector of the corresponding eigenvalues. The eigenvalues $\lambda_{i},\;\{i=1,2,\ldots ,d\}$ of the matrix *R* decrease in order and vary significantly at a particular value *K*, as signified in (8) and (9):(9)$$\lambda_{1}>\lambda_{2}>\cdots >\lambda_{k}>>\lambda_{k+1}>\cdots >\lambda_{d}$$(10)$$\Delta_{i}=\left\{\begin{array}{l} 1,\;i=1\\ \frac{\left|\lambda_{i}-\lambda_{i-1}\right|}{\lambda_{i-1}},\;i=2,3,\ldots ,d \end{array}\right.$$

When $\Delta_{i}$ is less than the set threshold for the first time, i is *K* at this point. Experimental observations demonstrate that the eigenvectors corresponding to smaller eigenvalues are mainly noise or irrelevant components after reconstruction, accounting for most *d* components. Therefore, the first *K* initial single component matrices are reconstructed to reduce algorithm time, giving the reconstruction formula as follows:(11)$$Z_{i}=Q_{i}Q_{i}^{T}X^{T},\;i=1,2,\ldots ,k.$$

By converting *K* initial single component matrix $Z_{i}$ into *K* initial symplectic geometric component $Y_{i}$ of length n, the diagonal averaging formula is obtained as follows:(12)$$q=\left\{\begin{array}{ll} \frac{1}{q}\sum_{p=1}^{q}z_{p,q-p+1}^{*} & \;1\leq q<d^{*}\\ \frac{1}{d^{*}}\sum_{p=1}^{d^{*}}z_{p,q-p+1}^{*} & d^{*}\leq q<m^{*}\\ \frac{1}{n-q+1}\sum_{p=q-m^{*}+1}^{n-m^{*}+1}z_{p,q-p+1}^{*} & m^{*}\leq q\leq n \end{array}\right.$$
where $y_{q}$ is the qth element of $Y_{i}$; $z_{f,g}$ indicates the element in the fth row and gth column of the matrix $Z_{i}$; $d^{*}=\min(m,d)$; $m^{*}=\max(m,d)$; and $z_{f,g}^{*}$ is given by:(13)$$z_{f,g}^{*}=\left\{\begin{array}{ll} z_{f,g} & \;\mathrm{if}\;m\leq d\\ z_{g,f} & \;\mathrm{if}\;m>d \end{array}\right.$$

Before recombining components, wavelet packet decomposition reconstruction is introduced for *K* initial symplectic geometric components $Y_{i}$ to enhance the algorithm’s noise suppression ability. Wavelet basis functions are selected to perform a three-layer wavelet packet decomposition on the initial symplectic geometric components. The noise content of these components may be decomposed into any wavelet packet coefficient. By calculating the sample entropy of all wavelet packet coefficients, we can assess the complexity of each coefficient; coefficients with higher complexity indicate greater noise content. We then eliminate the wavelet packet coefficient with the highest sample entropy and reconstruct the intermediate symplectic geometric component based on the processed wavelet packet coefficient.

For *K* intermediate symplectic geometric components, we employed the condensed hierarchical clustering method component recombination. At the beginning of clustering, each initial symplectic geometric component is treated as a cluster class $C_{i}$, and the distance between clusters $D(C_{i},C_{j})$ is obtained as follows:(14)$$D(C_{i},C_{j})=\frac{1}{n^{2}}\sum_{x\in C_{i}}\sum_{y\in C_{j}}\sqrt{{\sum_{p=1}^{n}\left|x_{p}-y_{p}\right|}^{2}}$$

We select the two clusters with the smallest distance among all available clusters. This process effectively combines a new cluster, recombining the corresponding two initial symplectic geometric components into one intermediate symplectic geometric component and merging them into k − 1 intermediate symplectic geometric components and k − 1 corresponding cluster classes. The termination condition judgment is obtained as follows:(15)$$R(Y_{i}^{\prime },Y_{j}^{\prime })<\theta \;i\neq j$$
where R(⋅) is the calculated correlation coefficient; $Y_{i}^{\prime },Y_{j}^{\prime }$ represents the intermediate symplectic geometric component in the clustering process; and θ is the termination threshold.

We continue the clustering and recombination operations until all components satisfy the termination condition, yielding multiple symplectic geometric wavelet components.

## 3. Prediction of the Time-Frequency Characteristics of Leakage Current Based on I-Informer

The electrical characteristic associated with the deterioration of the lightning arrester resistance plate manifests as changes in various components of the leakage current. By predicting the future trend of the component characteristics, we can effectively provide an early warning of the short plate of the lightning arrester. Traditional algorithms for long sequence prediction rely on multi-step forecasting. However, as the prediction length increases, cumulative errors tend to escalate. The Informer model is an improved Transformer model, addressing the issues of SeqToSeq long sequence prediction through encoding and decoding, thereby minimizing the effects of cumulative errors [20]. This article designs an I-Informer model to achieve the long-term prediction of leakage current characteristics, and the model’s overall structure is illustrated in Figure 1.

### 3.1. Model Input

Using the ISGMD-WP algorithm, we decompose the leakage current signal of the lightning arrester into several symplectic geometric components, each containing different harmonic components. The first three symplectic geometric components correspond to the original leakage current signal’s 6th, 12th, and 18th harmonic components. The mean of the original signal serves as the DC component feature of the leakage current. Moreover, we extracted the appropriate time-frequency features from the three harmonic components. Given their temporal correlation and robustness against noise, we screen for statistical features such as mean, average power, standard deviation, root square amplitude, peak value, and kurtosis. Previous studies show that these characteristics have good performance in the time and frequency domain, and they are strongly correlated with the deterioration of the resistor [16,17]. The root mean square amplitude of each harmonic component was selected as the representative feature. This is because it comprehensively represents the energy and has resilience to noise. This selection resulted in the final feature sequence of the signal, which includes four key attributes.

Although the Informer model significantly improves the speed of parallel data processing through its attention mechanism, it overlooks the sequential relationship that reflects the temporal characteristics of the input sequence. Thus, the input part needs to add position and time encoding. The position encoding employs sine encoding, while the time encoding considers the lightning arrester’s slow degradation speed and the long monitoring history cycle, using the four-timestamp format: hour, day, week, and month. Before inputting the feature sequence and the two types of encoding information into the encoder or decoder, data embedding integration is required using the following methods:(16)$${\mathrm{IN}}_{\mathrm{embed}}=Linear(Conv1d(x)⊕x_{pos}⊕Linear(x_{time}))$$
where x is the signal feature sequence; $x_{pos}$ indicates the sequence position encoding; $x_{time}$ refers to the sequence time encoding; Conv1d(⋅) is one-dimensional convolution; Linear(⋅) is fully connected; and ⊕ is the matrix concatenation operation.

### 3.2. Encoder

The encoder architecture of the model adopts a multi-branch hierarchical structure to progressively process the input sequence. Specifically, the first branch receives the complete input sequence, while each subsequent branch processes the latter half of the sequence from its preceding branch. Within each branch, the encoder is composed of a stack of multi-head probabilistic sparse self-attention layers and distillation layers. Notably, the number of stacked layers decreases by one as the input sequence length is halved in each subsequent branch, ensuring a hierarchical and efficient feature extraction process. This design enables the model to capture both global and local temporal dependencies while maintaining computational efficiency.

The multi-head probability sparse self-attention layer reduces the network’s time complexity while maintaining the attention mechanism’s feature extraction ability. Only a subset of all query vectors contributes significantly to the attention computation. Their sparsity is determined by calculating the attention distribution of each query vector and the KL divergence from a uniform distribution. After filtering the top n query vectors in descending order of evaluation values for self-attention distribution calculation, the attention distribution of the remaining query vectors is processed according to a uniform distribution. The implementation formula for multi-head probability sparse self-attention is obtained in (16):(17)$$A(Q,K,V)=\mathrm{Softmax}(\frac{\bar{Q}K^{T}}{\sqrt{d}})V$$
where $\bar{Q}$ contains the first n query vectors.

The distillation layer reduces the extracted feature sequence length by half, reducing the number of model parameters and the memory usage. This article introduces dilated convolution and residual connections in the distillation layer. The extended convolution can expand the receptive field without increasing the parameters, and it can capture the long-range time dependence, which is very important for modeling the slow degradation process. Residual connection can mitigate the information loss in the process of feature compression and ensures that the key patterns related to degradation are retained. The calculation formula is as follows:(18)$$X_{j+1}^{t}=\mathrm{MaxPool}(\mathrm{ELU}(Conv1d(X_{j}^{t})+DConv1d(X_{j}^{t})+X_{j}^{t}))$$
where DConv1d(⋅) is one-dimensional dilated convolution; ELU(⋅) represents exponential linear unit activation function; and MaxPool(⋅) signifies maximum pooling.

### 3.3. Decoder

The decoder adopts a non-autoregressive generative architecture to achieve one-step sequence prediction, eliminating the computational overhead and error accumulation of traditional step-by-step decoding. It comprises two key layers: (1) a masked multi-head probabilistic sparse self-attention layer that processes a concatenation of the encoder’s contextual features and a zero-initialized prediction placeholder, enforcing temporal causality through attention masking; and (2) a multi-head attention layer that performs feature fusion between the decoder’s intermediate representations and the encoder’s output, enabling cross-modal information integration while maintaining parallelizable computation. This design ensures efficient long-term dependency modeling without recursive prediction steps.

## 4. Experiments

The controllable self-recovery energy dissipation experimental device for converter stations is depicted in Figure 2a. The system has centralized control and a self-organizing local area network. It can achieve automatic control of the testing process and synchronize the comprehensive testing of multiple lightning arresters. The three-dimensional structure diagram is shown in Figure 2b. The arrester body is a 136-column series parallel arrester. Each column is composed of fixed elements and controlled elements, which are installed in groups. The three control switches include the trigger gap or thyristor switch K0; the fast mechanical switch K1, based on the electromagnetic repulsion operating mechanism; and the bypass switch K2, based on the hydraulic spring operating mechanism. In addition, the device also includes a voltage sharing resistance R1 for balancing the partial pressure ratio between the fixed element and the controlled element of the arrester.

### 4.1. Evaluation Effect of the Comprehensive Indicator

To assess the effectiveness of the proposed decomposition algorithm, we performed comparative experiments after centralizing the data. These experiments involve several decomposition methods: Complete Ensemble Empirical Mode Decomposition with Adaptive Noise (CEEMDAN), wavelet packet decomposition, Variational Mode Decomposition (VMD), and symplectic mode decomposition. Among them, the number of components K of VMD needs to be determined manually a priori. Therefore, the leakage current decomposition experiment will be carried out when the number of display components K is 3, 5 or 7. The decomposition results for leakage current produced by each algorithm are illustrated in Figure 3. As shown, the ISGMD-WP algorithm decomposes the leakage current into seven components, with the first three components being the prominent harmonics of the 12th, 18th, and 6th orders. The fourth, fifth, and sixth components represent additional harmonics; the residual components are noise. The main components of the leakage current can be extracted effectively.

Table 1 displays the experiment results of the ISGMD-WP algorithm using VMD and the original SGMD’s EIDC values after introducing Gaussian white noise with different signal-to-noise ratios to the simulated signal. The results indicate that the original SGMD algorithm has poor decomposition results. In contrast, the proposed ISGMD-WP significantly improves the decomposition ability of SGMD, with EIDC values lower than other comparison algorithms and outperforming the VMD algorithm. Notably, under substantial noise interference of −5 dB, the EIDC value of this algorithm is only 1.95, proving its good anti-noise ability.

### 4.2. Leakage Current Characteristic Prediction Experiment

The degradation process of the resistor of a lightning arrester is usually slow, and a shorter sampling interval can cause data redundancy. At the same time, fluctuations in the grid voltage at both ends of the controllable energy dissipation device can affect the accuracy of feature prediction. In this article, the sampling interval for long-term monitoring of leakage current data is 1 h, with a total duration of 120 days. Each dataset is extracted with four characteristics: DC mean, 6th harmonic root square amplitude, 12th harmonic root square amplitude, and 18th harmonic root square amplitude [21,22,23]. The history of the quantitative indicators for each feature is shown in Figure 4.

After the deterioration of the resistor, the DC mean characteristic changes significantly, while the three harmonic characteristics change minimally. Throughout the monitoring process, all four characteristics exhibit similar trend fluctuations with the fluctuation of the grid voltage. Therefore, this article will mainly predict the future changes in the DC mean characteristics to achieve short film warning. We conducted comparative prediction experiments to evaluate the effectiveness of the proposed prediction algorithms, including LSTM, GRU, and Informer networks. The dataset is divided into training and testing sets with an 80:20 ratio; the input step size is 96 and the output step size is 24. The batch size is 4, and the initial learning rate is 0.001.

As illustrated in Figure 5a, the predicted results of the four models are well aligned with the trend of actual values. Specifically, Figure 5a presents the total prediction results of each network with a step size of 24, and it is shown that the predicted results of the four models are consistent with the trend of actual values. While Figure 5b displays the residual results of the predicted signal and the actual signal. Among the models, the LSTM network demonstrated the least practical prediction effect, with numerous instances where the predicted values deviate significantly from the actual values, followed by GRU. Informer and I-Informer have predicted results closer to the actual values, with fewer significant deviations. To further evaluate the model’s predictive performance, experiments were conducted using the DC mean single feature prediction and four features to predict the DC mean single feature.

Table 2 displays the final results, demonstrating that the I-Informer model achieves better prediction outcomes than the other three comparative models, regardless of the number of input features used. Moreover, the prediction produced results with four features for forecasting the DC mean single feature, surpassing those obtained using the DC mean single feature to predict the single feature. This result highlights the value of incorporating extracted harmonic features as prediction inputs in the proposed methodology, ultimately enhancing the accuracy of the DC prediction.

## 5. Conclusions

A novel algorithm for decomposing leakage current signals has been developed, utilizing the ISGMD-WP. This algorithm integrates a short chip warning method based on the I-Informer network to predict component features. The symplectic geometric mode decomposition has been refined by optimizing the determination of the embedding dimension, the initial single-component matrix’s reconstruction, and the components’ recombination. Integrating wavelet packet decomposition reconstruction in the design of the ISGMD-WP algorithm significantly enhances the long-term predictive capabilities of the Informer network. This process is achieved by enhancing the input embedding layer and the encoding distillation layer. Experimental results demonstrate that the ISGMD-WP algorithm effectively decomposes leakage current, achieving an EIDC of 1.95 under strong noise conditions. The proposed I-Informer network performs better in long-term predictions, with MAE and RMSE values of 0.02538 and 0.03175, respectively, outperforming the comparative algorithm. Ultimately, this approach enhances the accurate prediction of the short chip state in lightning arresters based on forecast data.

## Figures and Tables

**Figure 1 sensors-25-03478-f001:**
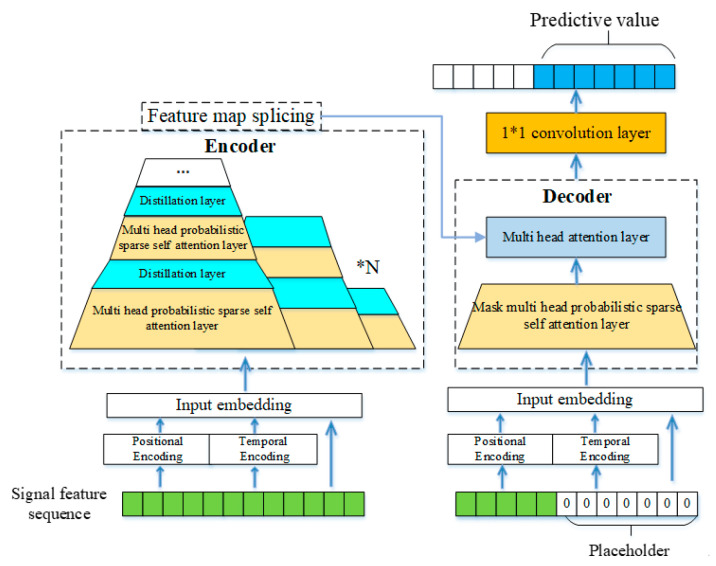
I-Informer model structure diagram.

**Figure 2 sensors-25-03478-f002:**
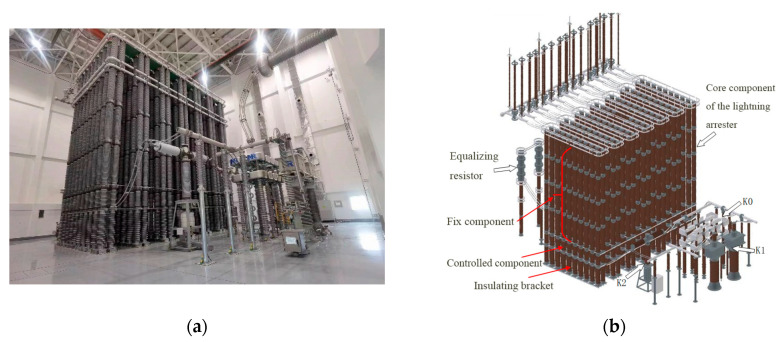
Controllable self-recovery energy dissipation experimental device (**a**) Experimental system (**b**) Three-dimensional model.

**Figure 3 sensors-25-03478-f003:**
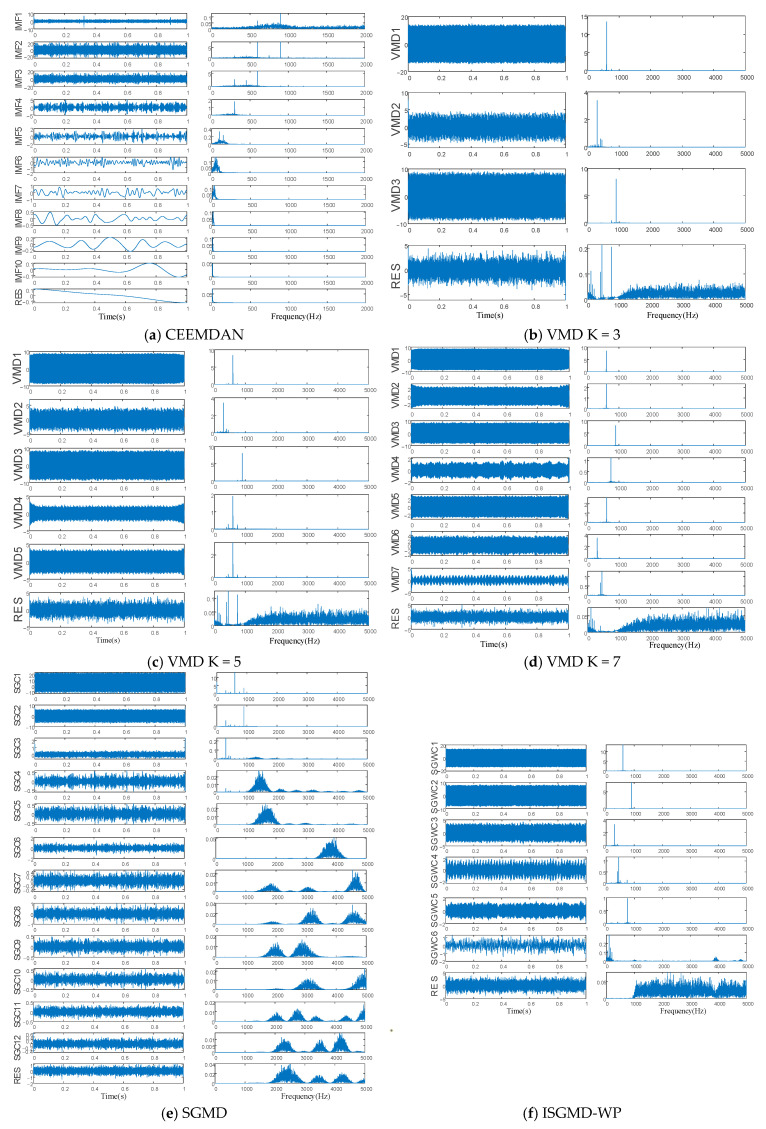
Time domain waveforms and spectrograms for different algorithms.

**Figure 4 sensors-25-03478-f004:**
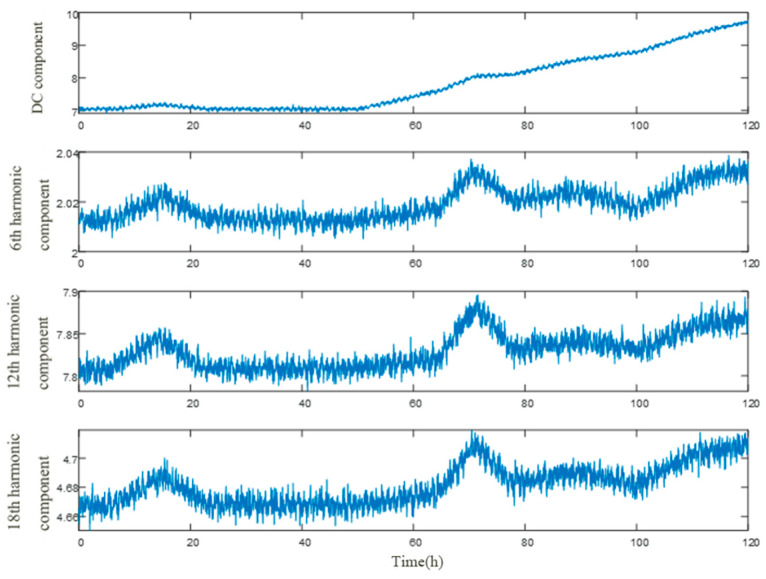
Long term characteristic curve of leakage current.

**Figure 5 sensors-25-03478-f005:**
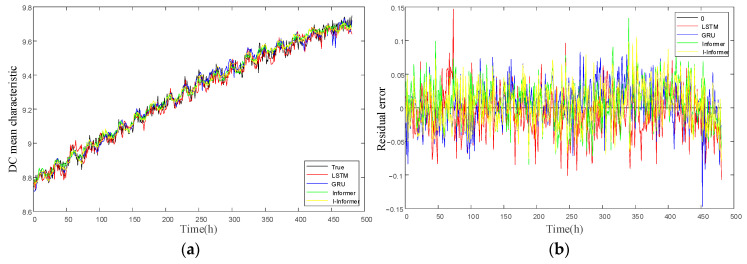
Comparison of Prediction Results of Various Models (**a**) Curve between predicted results and actual values (**b**) Residual curve.

**Table 1 sensors-25-03478-t001:** EIDC table for analog signals with different signal-to-noise ratios.

SNR	VMD	SGMD	ISGMD-WP
−5 dB	2.97597	3.26770	1.95689
5 dB	0.29769	0.67891	0.24502
15 dB	0.02907	0.44365	0.02264
25 dB	0.00511	0.42125	0.00203

**Table 2 sensors-25-03478-t002:** Statistical Table of Predictive Performance Indicators for Different Models.

Input Feature Count	Indicator Name	LSTM	GRU	Informer	I-Informer
1	MAE	0.0387477	0.0358729	0.0337567	0.0319757
	RMSE	0.0498476	0.0454695	0.0437456	0.0402823
4	MAE	0.0302019	0.0280184	0.0266243	0.0253844
	RMSE	0.0388004	0.0352935	0.0333787	0.0317544

## Data Availability

The data and code are considered intellectual property of the project and are therefore not publicly available.

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
