# Peer review of "A Degradation Warning Method for Ultra-High Voltage Energy Devices Based on Time-Frequency Feature Prediction"

_sensors, 2025, doi:10.3390/s25113478_

Round 1

Reviewer 1 Report

Comments and Suggestions for Authors

This study proposes a leakage current analysis methodology for ultra-high voltage energy devices, with experimental validation demonstrating the superior performance of the proposed strategy over conventional approaches. I have the following advices.

1. The manuscript requires several formatting adjustments to meet publication standards. In lines 140-153, the indentation and paragraph spacing should be standardized to ensure visual consistency throughout the document. Specific notation issues need attention, including the two "E1" labels in line 185 and inconsistent variable formatting in lines 186-188, where terms like d0 and i should be properly italicized. All mathematical equations should be center-aligned with their corresponding numbers right-aligned to maintain professional presentation standards.

2. The author contributions section at line 435 should be supplemented, which I believe is the format requirement of the journal.

3. Reference formatting requires careful standardization throughout the document.

4. Figure 2 should be more clear. The photo is vague.

5. Please verify whether units are specified for the y-axis in Figure 4 and supplement them if applicable.

Author Response

Dear,

Thank you very much for your letter dated May 19, 2025 and the attached comments from the associate editor and reviewers on the captioned paper.

First of all, we would like to give our great and sincere gratitude to you and the reviewers for the invaluable comments and constructive feedbacks on our previous submission.

In this revised submission, each comment or question by reviewers has been carefully addressed and explained. The attached note outlines the changes made in the revised version and our responses to each comment/question.

In addition, we also improved and refined the paper in general so that is clearer and more succinct to readers.

Thank you again for processing our paper and we look forward to hearing the decision on this submission soon.

Sincerely yours,

Lihui Wang

May 26 2025

In the following, we list all the concerns of each reviewer in italic font. It is immediately followed by how we revised the manuscript according to them (in regular font). The changes corresponding to the reviewers’ comments have been marked in red in the revised manuscript.

Response to Reviewer #1

Comments from Reviewer #1:

[Comment 1]

The manuscript requires several formatting adjustments to meet publication standards. In lines 140-153, the indentation and paragraph spacing should be standardized to ensure visual consistency throughout the document. Specific notation issues need attention, including the two "E1" labels in line 185 and inconsistent variable formatting in lines 186-188, where terms like d0 and i should be properly italicized. All mathematical equations should be center-aligned with their corresponding numbers right-aligned to maintain professional presentation standards

Response:

This suggestion is very helpful to us. We have made modifications to the format of the manuscript. Among them, the format in lines 140-153 remains consistent with the previous text. The formulas and variable formats in lines 185-188 have been modified. In addition, all formulas are kept in a centered format with right aligned numbering.

[Comment 2]

The author contributions section at line 435 should be supplemented, which I believe is the format requirement of the journal.

Response:

Thanks for your question. We have added the section contributed by the author.

[Comment 3]

Reference formatting requires careful standardization throughout the document.

Response:

Thanks for your question. We have revised the format of the references according to the requirements of the journal.

[Comment 4]

Figure 2 should be more clear. The photo is vague.

Response:

Thank you for your question. We have replaced Figure 2 with a clearer image.

[Comment 5]

Please verify whether units are specified for the y-axis in Figure 4 and supplement them if applicable.

Response:

Thank you for your suggestion. After our confirmation, the vertical axis in Figure 4 represents the measurement indicators of different component features, so there is no unit. We did not express it clearly in the text before, so we made modifications to the image description section.

Reviewer 2 Report

Comments and Suggestions for Authors

SUMMARY

The article presents a novel method for enhancing the decomposition and analysis of leakage current signals in converter station lightning arresters, which are crucial for evaluating the health and performance of power system insulation. At the core of their research, the authors propose an improved signal decomposition method, ISGMD-WP, which combines Improved Symplectic Geometry Mode Decomposition with Wavelet Packet analysis. To evaluate the performance of the proposed method, they conducted comparative experiments using several decomposition techniques, including CEEMDAN, VMD, wavelet packet decomposition, and the original SGMD. Through extensive testing with both simulated signals and the introduction of Gaussian white noise at varying levels, the study demonstrates that ISGMD-WP outperforms the other methods in terms of decomposition accuracy and noise resistance.

POSITIVE ASPECTS

1. Based on the literature review, the authors conducted a comprehensive literature review and identified key findings and gaps in current knowledge and practices regarding lightning arrester monitoring in DC power systems.
2. The authors developed and applied an improved leakage current harmonic separation algorithm to analyze lightning arrester signals more accurately and extract fault-relevant components.
3. The authors designed a predictive model called the I-Informer to forecast the future time-frequency characteristics of the leakage current, which are indicators of lightning arrester degradation.

CONCERNS

The presented work is useful but has some concerns that need to be removed. Points that must be addressed by authors are listed below:

Major concerns
1. The left side of equation (12) is unclear.
2. The texts used in Fig. 2b are not comfortably readable. Use a bigger font in the texts in Fig. 2b.
3. Please add several references to the manuscript that confirm the 6th, 12th, and 18th harmonic components of the leakage current as a significant indicator of the aging of lightning arresters.

Minor concerns
1. The letter T, denoting a matrix transformation, is written in normal font.
2. The numerical value of the physical unit and its symbol shall be separated by a space according to ISO 80000-1: 2009 standard (e.g. line 370). Correct all text accordingly including tables.
3. In scientific articles, a forward slash (solidus "/") followed by a physical unit symbol is not usually used. The symbol of the physical unit should be enclosed in parentheses, separated from the text by a space. Correct all text accordingly, including figures.

REMARKS

1. In Fig. 2b, it would be appropriate to label the time domain axes and the spectrograms.

QUESTIONS

I have four questions for the authors of the article.

1. Why did the authors introduce Gaussian white noise with different signal-to-noise ratios into the simulated signal?
2. Does the Gaussian white noise introduction realistically reflect conditions on surge arresters?
3. Are there other types of noise that would better reflect the operation in electrical power networks?
4. Various scientific articles confirm the third harmonic component of the leakage current (or the fifth harmonic component) as a significant aging (degradation) indicator of lightning arresters. Why did the authors use the amplitude of the 6th, 12th, and 18th harmonics for extracting?

Answer the given questions with comments in the manuscript.

CONCLUSION

I find this article helpful. Regretfully, the paper cannot be accepted in its present form. The authors of the present article have to correct the issues.

Author Response

Dear,

Thank you very much for your letter dated May 19, 2025 and the attached comments from the associate editor and reviewers on the captioned paper.

First of all, we would like to give our great and sincere gratitude to you and the reviewers for the invaluable comments and constructive feedbacks on our previous submission.

In this revised submission, each comment or question by reviewers has been carefully addressed and explained. The attached note outlines the changes made in the revised version and our responses to each comment/question.

In addition, we also improved and refined the paper in general so that is clearer and more succinct to readers.

Thank you again for processing our paper and we look forward to hearing the decision on this submission soon.

Sincerely yours,

Lihui Wang

May 26 2025

In the following, we list all the concerns of each reviewer in italic font. It is immediately followed by how we revised the manuscript according to them (in regular font). The changes corresponding to the reviewers’ comments have been marked in red in the revised manuscript.

Response to Reviewer #2

Comments from Reviewer #2:

Major concerns:

[Comment 1]

The left side of equation (12) is unclear.

Response:

Thanks for your question. The left side of equation 12 was mistakenly written in subscript form, but it has now been corrected to its correct form.

[Comment 2]

The texts used in Fig. 2b are not comfortably readable. Use a bigger font in the texts in Fig. 2b

Response:

Thanks for your question. We have enlarged the font in Figure 2 (b).

[Comment 3]

Please add several references to the manuscript that confirm the 6th, 12th, and 18th harmonic components of the leakage current as a significant indicator of the aging of lightning arresters.

Response:

Thanks for your question. We have added 3 references, numbered 21-23.

Minor concerns:

[Comment 1]

The letter T, denoting a matrix transformation, is written in normal font.

Response:

Thanks for your question. We have modified T in matrix operations to a regular font.

[Comment 2]

The numerical value of the physical unit and its symbol shall be separated by a space according to ISO 80000-1: 2009 standard (e.g. line 370). Correct all text accordingly including tables.

Response:

Thanks for your question. We have added spaces before physical symbols, especially dB, in all texts and tables to comply with the ISO 80000-1: 2009 standard.

[Comment 3]

In scientific articles, a forward slash (solidus "/") followed by a physical unit symbol is not usually used. The symbol of the physical unit should be enclosed in parentheses, separated from the text by a space. Correct all text accordingly, including figures.

Response:

Thanks for your question. We will modify all occurrences of "/physical unit symbol" in text and charts to "(physical unit symbol)".

Questions:

[Comment 1]

Why did the authors introduce Gaussian white noise with different signal-to-noise ratios into the simulated signal?

Response:

Thanks for your question. We introduce Gaussian white noise with different signal-to-noise ratios into the simulated signal, and by constructing test signals containing different noise energies, we can quantitatively evaluate the algorithm's ability to suppress mode aliasing. In addition, the coupling relationship between the nonlinear characteristics of leakage current signals in power equipment and noise interference can be reflected through white noise modeling, which can more comprehensively evaluate the robustness and effectiveness of algorithms in practical applications where noise is an inevitable factor.

[Comment 2]

Does the Gaussian white noise introduction realistically reflect conditions on surge arresters?

Response:

Thanks for your question. We introduce Gaussian white noise with different signal-to-noise ratios into the simulated signal to test and evaluate the performance of the ISGMD-WP algorithm under real-world conditions that surge arresters may encounter. In practical scenarios, surge arresters operate in environments where noise is an inevitable factor, such as electromagnetic interference, thermal noise, and other background disturbances. By introducing Gaussian white noise, the flat spectral characteristics of white noise can simulate wideband electromagnetic interference in the converter station environment (such as IGBT switch noise, random discharge and other composite noise superposition), thus evaluating the ability of ISGMD-WP algorithm to decompose and extract useful features from leakage current signals in the presence of different noise levels. This evaluation helps to determine the robustness and reliability of the algorithm in real-world applications where noise cannot be completely eliminated. The results of these tests indicate that even under significant noise interference, the ISGMD-WP algorithm outperforms other algorithms, providing evidence for the practical effectiveness of surge arrester monitoring and alarm systems..

[Comment 3]

Are there other types of noise that would better reflect the operation in electrical power networks?

Response:

Thanks for your question. The selection of Gaussian white noise is mainly to provide a controlled and standardized method for evaluating the anti noise performance of algorithms. However, in actual power networks, other types of noise such as electromagnetic interference (EMI), power line noise, and impulse noise may also be commonly present. These noises may have different characteristics and may more accurately reflect the real working environment of surge arresters. We fully understand that complex forms of interference such as pulse type and colored noise may exist in the power system, and their statistical characteristics differ significantly from Gaussian white noise. The Gaussian noise model used in this study is mainly based on two considerations: firstly, in benchmark studies lacking clear scene priors, the Gaussian model can provide an interpretable theoretical framework for system performance analysis, such as the linear superposition characteristics of signals and noise, which facilitate the derivation of sensitivity functions; Secondly, the existing industry standards commonly use this model as a basic reference in electromagnetic compatibility assessment. Based on preliminary analysis of actual scenarios, the current model demonstrates sufficient applicability under typical operating conditions. Your suggestion is highly forward-looking. We will systematically study the robustness improvement methods for non Gaussian scenes in future work by constructing an extended dataset containing multidimensional noise features.

[Comment 4]

Various scientific articles confirm the third harmonic component of the leakage current (or the fifth harmonic component) as a significant aging (degradation) indicator of lightning arresters. Why did the authors use the amplitude of the 6th, 12th, and 18th harmonics for extracting?

Response:

Thanks for your question. We believe that, unlike the 3rd and 5th harmonics, the selection of the 6th, 12th, and 18th harmonics is mainly based on two considerations: firstly, the measurement characteristics of the experimental system result in the effective enhancement of high-frequency harmonic signals, and the response characteristics of sensors in specific frequency bands give higher-order harmonics (such as ≥ 6th harmonics) a higher signal-to-noise ratio advantage in detection; Secondly, there is a direct correlation between the microphysical mechanism caused by early aging of lightning arresters and the 6n harmonic spectrum, which can avoid the problem of insufficient sensitivity of traditional low-frequency harmonic analysis to gradual failure. This method selection reflects the specific adaptability between the detection accuracy requirements and the device degradation mechanism. In addition, reference 22 also studied the influence of high-order harmonics on the degradation of lightning arresters.

Round 2

Reviewer 2 Report

Comments and Suggestions for Authors

SUMMARY

The article presents a novel method for enhancing the decomposition and analysis of leakage current signals in converter station lightning arresters, which are crucial for evaluating the health and performance of power system insulation. At the core of their research, the authors propose an improved signal decomposition method, ISGMD-WP, which combines Improved Symplectic Geometry Mode Decomposition with Wavelet Packet analysis. To evaluate the performance of the proposed method, they conducted comparative experiments using several decomposition techniques, including CEEMDAN, VMD, wavelet packet decomposition, and the original SGMD. Through extensive testing with both simulated signals and the introduction of Gaussian white noise at varying levels, the study demonstrates that ISGMD-WP outperforms the other methods in terms of decomposition accuracy and noise resistance.

POSITIVE ASPECTS

1. Based on the literature review, the authors conducted a comprehensive literature review and identified key findings and gaps in current knowledge and practices regarding lightning arrester monitoring in DC power systems.
2. The authors developed and applied an improved leakage current harmonic separation algorithm to analyze lightning arrester signals more accurately and extract fault-relevant components.
3. The authors designed a predictive model called the I-Informer to forecast the future time-frequency characteristics of the leakage current, which are indicators of lightning arrester degradation.

CONCERNS

The presented work is useful but has some concerns that need to be removed. Points that must be addressed by authors are listed below:

Minor concerns
1. The word 1 Introduction is redundantly included in the keywords.
2. The letter T, denoting a matrix transformation, is written in normal font see Eq.(11).
3. In Fig. 3, the time and frequency axes are missing.

CONCLUSION

I find this article helpful. Regretfully, the paper cannot be accepted in its present form. The authors of the present article have to correct the issues.

Author Response

Cover letter

Title:A Degradation Warning Method for Ultra-high Voltage Energy Devices Based on Time-frequency Feature Prediction

Authors:Pin-Zhang Zhao, Li-Hui Wang, Jian Wei, Yi-Fan Wang, Hai-Feng Wu

Publication: Sensors

Dear editor,

Thank you very much for your letter dated May 29, 2025 and the attached comments from the associate editor and reviewers on the captioned paper.

First of all, we would like to give our great and sincere gratitude to you and the reviewers for the invaluable comments and constructive feedbacks on our previous submission.

In this revised submission, each comment or question by reviewers has been carefully addressed and explained. All updated sections are highlighted in the manuscript.

The attached note outlines the changes made in the revised version and our responses to each comment/question.

In addition, we also improved and refined the paper in general so that is clearer and more succinct to readers.

Thank you again for processing our paper and we look forward to hearing the decision on this submission soon.

Sincerely yours,

Lihui Wang

May 29 2025

In the following, we list all the concerns of each reviewer in italic font. It is immediately followed by how we revised the manuscript according to them (in regular font). The changes corresponding to the reviewers’ comments have been marked in red in the revised manuscript.

Response to Reviewer #2

Comments from Reviewer #2 Round #2:

[Comment 1]

The word 1 Introduction is redundantly included in the keywords.

Response:

This suggestion is very helpful to us. We have removed duplicate introductions.

[Comment 2]

The letter T, denoting a matrix transformation, is written in normal font see Eq.(11).

Response:

Thanks for your question. We have made modifications before, but not completely. In this modification, we conducted a detailed check and changed all T representing matrix operations to regular font. Include Eq.(11). And Eq.(16).

[Comment 3]

In Fig. 3, the time and frequency axes are missing.

Response:

Thanks for your question. We have modified the six subgraphs in Figure 3 by adding a time axis (Time/s) and a frequency axis (Frequency/Hz) to them.